https://doi.org/10.1038/s41467-022-30377-6　　**OPEN**

# Dynamic recognition and mirage using neuro-metamaterials

Chao Qian [1,2,3 ✉], Zhedong Wang[1,2,3], Haoliang Qian [1,2,3], Tong Cai[1,2,3], Bin Zheng [1,2,3], Xiao Lin[1,2,3], Yichen Shen [4], Ido Kaminer [5], Erping Li [1,2,3 ✉] & Hongsheng Chen [1,2,3 ✉]

Breakthroughs in the field of object recognition facilitate ubiquitous applications in the modern world, ranging from security and surveillance equipment to accessibility devices for the visually impaired. Recently-emerged optical computing provides a fundamentally new computing modality to accelerate its solution with photons; however, it still necessitates digital processing for in situ application, inextricably tied to Moore's law. Here, from an entirely optical perspective, we introduce the concept of neuro-metamaterials that can be applied to realize a dynamic object- recognition system. The neuro-metamaterials are fabricated from inhomogeneous metamaterials or transmission metasurfaces, and optimized using, such as topology optimization and deep learning. We demonstrate the concept in experiments where living rabbits play freely in front of the neuro-metamaterials, which enable to perceive in light speed the rabbits' representative postures. Furthermore, we show how this capability enables a new physical mechanism for creating dynamic optical mirages, through which a sequence of rabbit movements is converted into a holographic video of a different animal. Our work provides deep insight into how metamaterials could facilitate a myriad of in situ applications, such as illusive cloaking and speed-of-light information display, processing, and encryption, possibly ushering in an "Optical Internet of Things" era.

[1] ZJU-UIUC Institute, Interdisciplinary Center for Quantum Information, State Key Laboratory of Modern Optical Instrumentation, Zhejiang University, Hangzhou 310027, China. [2] ZJU-Hangzhou Global Science and Technology Innovation Center, Key Lab. of Advanced Micro/Nano Electronic Devices & Smart Systems of Zhejiang, Zhejiang University, Hangzhou 310027, China. [3] Jinhua Institute of Zhejiang University, Zhejiang University, Jinhua 321099, China. [4] Lightelligence Inc., Boston, MA 02210, USA. [5] Department of Electrical and Computer Engineering, Technion–Israel Institute of Technology, Haifa 32000, Israel. ✉email: chaoq@intl.zju.edu.cn; liep@zju.edu.cn; hansomchen@zju.edu.cn

Object recognition is a computer technology that involves computer vision and image processing to detect, classify, and tag instances of semantic objects of a certain class in digital photographs[1]. In modern society, it has already permeated deeply into every corner of our life, such as through video surveillance, target tracking, and image annotation and segmentation. To achieve object recognition, conventional approaches typically entail a two-step procedure: image sequences are captured by a camera and then processed using a digital computer, in tandem with deep learning or other pattern recognition algorithms[2,3]. However, with the exponential growth of Big Data and the Internet of Things, the conventional two-step procedure may not be considered adequate, with speed-of-light parallel information collection and data processing being in high demand.

From the beginning of this century, the advent of metamaterials and photonics has extensively motivated scientists to revisit the established object-recognition technology from a radically new perspective of optics. As compared with conventional electron-based implementations, those of photon-based computing have been found promising, especially in high-throughput, site-specific, real-time tasks, and provide the competitive advantages of speed-of-light operations, low-power consumption, and parallel capability[4]. Thus far, achievements include mathematical operators[5,6], logic operators[7,8], and so on. With respect to optical imaging, there exist also many related developments, such as computational imaging[9–11], edge detection[12,13], and non-line-of-sight imaging[14]. These pioneering approaches have effectively accomplished the first step of object recognition—image capture—replacing and even improving digital camera technology. However, if one intends to continue to the image processing step of an object-recognition task, digital computing remains indispensable for the required operations, including dimension reduction and feature extraction, and researchers are aiming to improve the efficiency of algorithms for complicated applications scenarios.

On the other hand, whereas object recognition (e.g., of handwritten digits) has been frequently used as an example to demonstrate various optical neural network architectures[15–20], such as nanophotonic deep-learning circuits and hybrid optical-electronic convolutional neural networks, these architectures in essence finalize the second step—image processing—replacing and even improving on digital computing by utilizing a series of previously prepared datasets. As such, to cater to in situ applications, a light-to-electronics/electronics-to-light conversion should be applied to allow real-time communication between the electronic image perceptron and optical processing component. The speed of the entire recognition system is ultimately limited by the cloak rate of an electronic processor[4]. The processing speed is difficult to increase due to the energy consumption of the processor, becoming ever more challenging at the twilight of Moore's law[21]. Given all these factors, it is of prime importance, albeit challenging, to facilitate entirely optical object recognition for real-world three-dimensional (3D) applications and thus harness the full potential of optical technology.

In this article, we propose a concept of neuro-metamaterials to realize direct and dynamic 3D object recognition. Neuro-metamaterials are capable of translating user-oriented demands into the structure of the spatial metamaterial by optimization techniques, and thus can automatically analyze or process scattered waves (we term it as a scattering neural network). In general, neuro-metamaterials can be embodied in either passive (inherent intelligence) or active (external-driven intelligence) forms. We show a proof of concept involving a living rabbit that plays freely in front of a passive neuro-metamaterial. The spatial electromagnetic (EM) fields scattered by the rabbit are re-scattered by the metamaterials, which process them to represent the rabbit's postures directly. After the neural network has been trained, the recognition accuracy rate reaches 98% in simulation for representative rabbit postures, including walking, standing, and sitting. The capability we developed also enables a counter-intuitive dynamic optical mirage to be created, as an example of which we converted a sequence of rabbit movements into a holographic video of a giraffe. Our work brings closer the implementation of state-of-the-art entirely photon-based computing in real-world applications and simplifies previously established but difficult-to-realize physical concepts. Looking ahead, the results of this study open a pathway for the development of this technology for a broad range of applications, such as intelligent metasurfaces-aided wireless communication[22], illusive cloaking[23], and information processing and display[24,25], together creating a vision of exciting possibilities in an "Optical Internet of Things".

## Results

**Prospective application scenario and the physical mechanism of neuro-metamaterials.** The potential applications of neuro-metamaterials are many and varied. Here, we take a security inspection system as an example of these promising applications and clarify the physical mechanism, as visualized in Fig. 1. Conventionally, when passengers are about to use public transport, they must always undergo screening by a hand-held metal detector or an advanced millimeter-wave scanner; meanwhile, their baggage is separately scanned by an explosive materials detection system or an x-ray system. In our vision, a passive "neuro-metamaterial wall," composed of layered or layer-free inhomogeneous metastructures, can also execute the detection and recognition task. When a man stands in front of the wall under the EM wave illumination, the security inspection results will automatically display. The underlying physical mechanism can be understood as follows. Since microwaves can readily penetrate most clothing material, a man carrying prohibited items

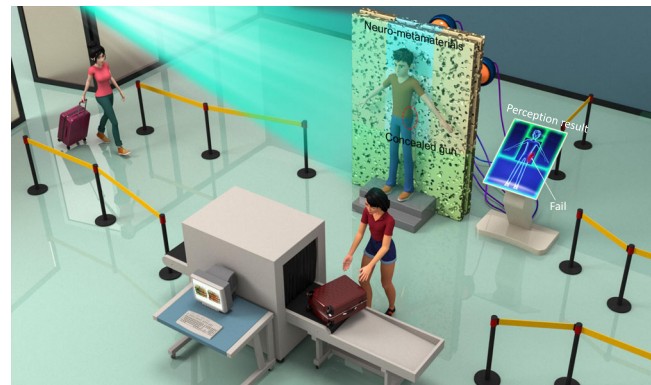

**Fig. 1 Illustrative usage of neuro-metamaterials in a security inspection system.** When a man goes through the security inspection system, he stands directly in front of a passive "neuro-metamaterial wall" fabricated from inhomogeneous metastructures. Under EM wave illumination, the wall performs the entire image processing and provides the result of the inspection in the form of an EM signal. Thus, the entire processing occurs without the need to convert the photonic to the (slower) electronic signal. In this example, to mitigate the complexity of designing the "neuro-metamaterial wall," we deliberately divide it into several subregions represented by different colors, each of which is responsible for one specific part of the human body. For example, a gun hidden at the waist will induce distinct scattered waves that will propagate and scatter further inside the wall and are translated directly to a warning signal at the customer interface. The figure was created by using Autodesk Maya and Photoshop software.

(such as concealed weapons) will induce strong and distinct EM scattered waves[26]; the waves will then propagate through the wall, within which intricate diffractions occur, and finally will be focused on different pre-defined locations behind the wall. Mathematically, the above process can be expressed as $U^{l+1}=\mathcal{F}^{-1}\left(\mathcal{F}\left(S_l \cdot U^l\right) \cdot H_l\right)$, where $H_l$ is the transfer function of free-space addressed by Rayleigh–Sommerfeld diffraction law, $\mathcal{F}$ denotes the Fourier transform operator, and $S_l$ is the scattering matrix[27,28]. Note that the value of $l$ ranges from 1 to $L$; $L$ is the number of metamaterial layers. $U^1$ is the initial EM field induced by passenger and $U^{L+1}$ is the EM field at the output plane. Our goal is to minimize the loss function, $f = \frac{1}{N}\sum_{n=1}^{N}||U_n^{L+1}|^2 - |G_n^{L+1}|^2|$, where $N$ is the number of samples and $G$ is the ground truth. To achieve this goal, we can utilize gradient-based approaches, such as topology optimization[29], or evolutionary approaches, such as simulated annealing algorithm[30], to optimize the scattering matrix.

To mitigate the design complexity, we deliberately divide the "neuro-metamaterial wall" into several subregions represented by different colors, as shown in Fig. 1, each of which is responsible for one specific part of the human body, in analogy with conventional safety check door. According to the EM strengths at different pixels, we can quickly evaluate the level of danger. For example, if the man is carrying a gun at his waist, the computer interface will show a warning signal for security officers. Compared with electronic security systems (mostly based on amplitude imaging technology)[31], neuro-metamaterials may find the advantages by directly and simultaneously utilizing multiply scattered wave properties, including phase, amplitude, and polarization, to reveal features that are otherwise invisible[32]. Such neuro-metamaterials-enabled object recognition will be more convenient and immediate and can be readily extended to other applications, such as intelligent prefilter in astronomy and driverless vehicles and autonomous drones[33].

**Neuro-metamaterial design and its numerical simulations.** Specifically, we chose a tame animal, a rabbit, as the research subject because of its biological similarities (body structure, EM responsiveness, and other congruences) with human beings[34], and we aimed to identify its postures. Thus, the subsequent tasks included mainly data collection, scattering matrix optimization, and neuro-metamaterials design. For data collection, we considered three sizes of rabbit (large, medium, and small), three postures (standing, sitting, and walking), the angles with respect to the normal of the neuro-metamaterials (from −30° to 30° with steps of 10°), and the distances from the rabbit to the neuro-metamaterials (from 10 mm to 100 mm with steps of 10 mm); see details in Supplementary Fig. 1. Using the commercial software package CST Microwave Studio, we simulated all these situations and created a database containing hundreds of samples.

For scattering matrix optimization, we deployed a diffractive neural network algorithm for simplicity (other algorithms, such as topology optimization and genetic algorithms, are also superior candidates), in analogy with orthodox artificial neural networks[16]. Each hidden layer of the network consisted of $30\times 40$ neurons, the physical dimension of which was $13\times 13$ mm$^2$, and the axial distance between layers was set to 300 mm (a hyperparameter that can be optimized) working at 8.6 GHz. The simulated data were shuffled and then fed to the neural network to accelerate the convergence of the algorithm. Among these data, 80% were used to generate the gradient, and the remaining 20% for testing. To facilitate its physical implementation, we assumed the transmitted amplitude of the neuron was uniform. Figure 2c plots the training process, where the loss declines significantly, and ultimately the

accuracy rates for the training and test set reach 99% and 98%, respectively. The close accuracy rates indicate that the neural network is reliable with little overfitting, which has also been checked by cross-validation (Methods). The final obtained transmitted phases are shown on the left in Supplementary Fig. 3. We blindly selected several samples from the testing set and show their numerical simulation results in Fig. 2d. Here, we designate three small regions with a radius of less than one wavelength that corresponds respectively to the three postures. The classification criterion of the posture is the single-pixel detector (left/middle/right) with the maximum signal. Evidently, all the focusing points are located at the correct position in accordance with the training results.

We then progressed to the design of the neuro-metamaterials. Many neuro-metamaterials can be realized, such as inhomogeneous bulky metamaterials and transmission metasurfaces (regarded as layered metamaterials). For conceptual clarity, we considered two-layer neuro-metamaterials. Neuro-metamaterials are constructed from a dense array of subwavelength meta-atoms, each of which functions as an independent neuron and interconnects to other meta-atoms of the subsequent layers. This design principle and pre-trained phase masks can be scaled into other single frequencies. For different frequencies, we should design specific metamaterials due to material dispersions to match the pre-trained phase masks, such as TiO$_2$ metasurfaces in visible[35]. At the microwave regime, we designed a metallic layered composite unit cell, the transmitted phase of which covers almost $2\pi$. However, we would like to note that, since the transmitted phase changes very sharply and the amplitude declines dramatically in the highlighted region (bottom right in Fig. 2b), they will be very sensitive to a slight variation in the metallic patch geometry and dielectric constant of the substrate in the fabrication. Although this problem can be alleviated to a certain extent by using a larger number of stacked layers, this would degrade the robustness of the neural network and induce strong mutual coupling[36]. Thus, we utilize only the working regime outside the highlighted region in the dispersion relation.

**Experimental results.** The experimental setup is delineated in Supplementary Fig. 4. A rabbit plays freely in front of the neuro-metamaterials under the illumination of a transverse electric plane wave, where the electric field is along the $z$ axis. Meanwhile, three detectors (small monopole antennas) are connected to a series of miscellaneous components to perceive the EM strength (the amplitude of the electric field) at the output plane[37]; see Methods and Supplementary Note 4. Note that here all the electronic components are used only to illustrate the perception results for users utilizing microwave frequencies, whereas in visible frequencies the results can be seen with the naked eye.

We conducted many experiments using two rabbits (one large gray rabbit and one small black-white rabbit); the results are presented in Fig. 3. To quantize the measurement results, we define an indicator—the variance in the amplitude of the electric field over its original amplitude, that is, $\triangle E/E_0$. Figure 3a shows photographs of all six postures (three per rabbit), corresponding one-to-one to the average $\triangle E/E_o$ of many measurements in Fig. 3b. Evidently, the strongest signal occurs at the expected locations, albeit only with a slight transcendence for a few postures. On this foundation, we further trained the two rabbits to continue to play for a period of time and recorded the time-varying signals, as delineated in Fig. 3c, 3d; see full dynamics in Supplementary Movies 1 and 2. Interestingly, in the first video, the large gray rabbit is very active, affording a dramatic variation in the signal (consistent with the ground truth). In contrast, in the second video, the small black-white rabbit (~5 months old) is not

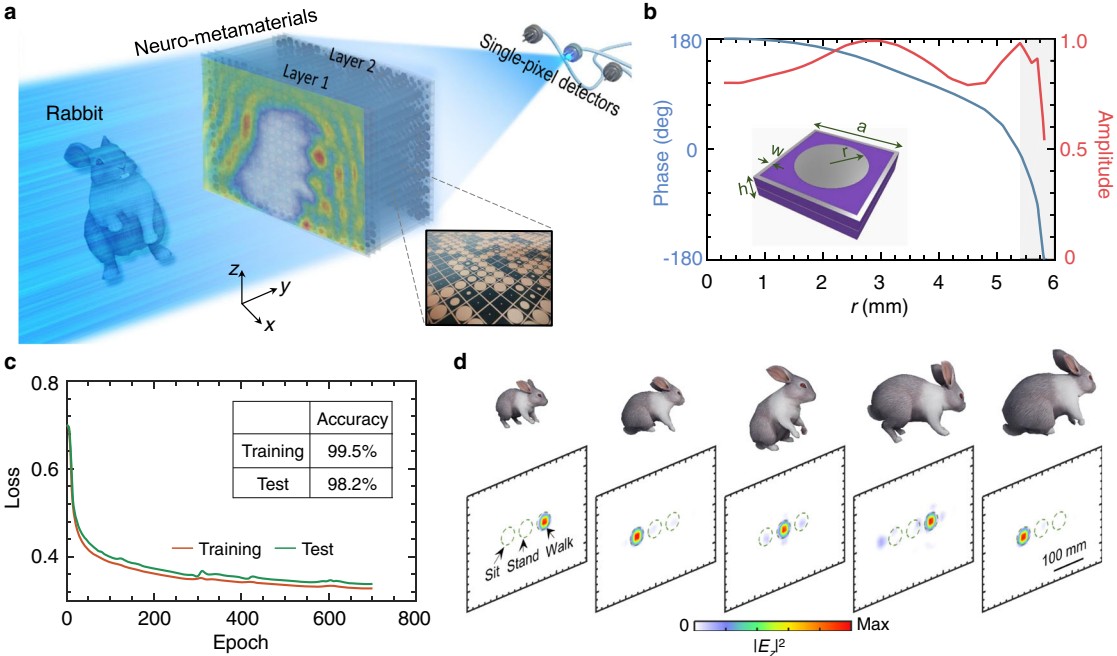

**Fig. 2 Design of neuro-metamaterials for recognition of a rabbit's postures. a** A rabbit freely plays in front of the specially-designed neuro-metamaterial, the scattered waves of which focus on different pre-defined pixels; see experimental setup in Supplementary Note 3. **b** Dispersion relation of the neuro-metamaterials. We deploy a polarization-insensitive composite unit cell patterned with circular patches; the surrounding square ring can reduce multiple coupling effects[36]. The detailed geometrical parameters are $a = 13mm, w = 0.5mm, h = 3mm$, and the relative permittivity of the F4B substrate is 2.65. **c** The loss of the training and test sets vs. epoch. After hundreds of epochs, the neuro-metamaterials achieve an accuracy rate of 99.5% and 98.2% for the training and test set, respectively. **d** Numerical simulations of five samples from the test set. In these samples, the rabbits differ in their size, posture, rotation angle, and distance to the neuro-metamaterials; see the inset pictures. Here, we judiciously define the optical recognition result (sit/stand/walk) to focus on one of the three positions (left/middle/right).

active, and thus, the signal remains relatively stable. Although the sitting and walking postures of the small rabbit are somewhat similar, our experimental results can express this hybrid mode in the second half of the regime (right panel in Fig. 3d).

**Dynamic optical mirage.** In contrast to object recognition, a counterintuitive optical phenomenon, a dynamic optical mirage/illusion, can be enabled by neuro-metamaterials. The realization of optical illusions using transformation optics was theoretically proposed in 2009, but experimentally it is hindered to a great extent by the complicated composition of metamaterials, which feature both anisotropy and inhomogeneity[23]. Our proposed approach may provide a radically new methodology that can simplify the pioneering scheme by virtue of two aspects. First, our design is more accessible for practical execution using neuro-metamaterials (isotropic) or metasurfaces. Second, our design can operate not only for static but also for non-static objects. As shown in Fig. 4a, when a rabbit plays freely in front of the neuro-metamaterial wall, an observer behind the wall will misconstrue it as another object, such as a giraffe. This scheme may enrich optical holograms, giving rise to many potential applications, such as data encryption, optical display, and illusive cloaking.

To demonstrate the above capability, we continuously extracted 15 image frames from Supplementary Movie 1 as the input and a giraffe video (which also contains 15 frames) was the output. Notice that optical mirage here is treated as an optimization task, rather than an inference task (which has been verified in the posture recognition experiment). To facilitate the neural network training, the raw input and output images were spatially sampled by $70 \times 84$ points, and the distance between points is set to be 13 mm (the size of the neuron); thus, the physical sizes of the input and output are $910 \times 1092$ mm$^2$. Input

images are encoded into the amplitude of the input field, and the neuro-metamaterials are optimized to transform rabbit images into giraffe images. We considered phase-modulated neuro-metamaterials, and the incident wavelength was set to be the same as that in Fig. 2. Figure 4b, c show the results, and Supplementary Movie 3 provides a full animation. The original rabbit movements are successfully reconstructed to provide a dynamic mirage of a drinking giraffe, with its head going down gradually. To quantitatively characterize the mirage performance, we adopted a structural similarity (SSIM) index with an average value of 93.51%[38].

In experiment, we take four distinct frames (frames 1, 7, 10, 15) as examples. For simplicity, the frames are resized into $325 \times 360$ mm$^2$ amplitude's modulated plates as the input of neuro-metamaterials (Fig. 5a). We fabricated the pre-trained neuro-metamaterials and measured the electric field distributions at the output plane. The measured results are displayed in Fig. 5b, which clearly shows a dynamic mirage of giraffe, and the giraffe's head goes down gradually. For comparison, we also measured the electric field distributions without the neuro-metamaterials, which shows a completely different result (Fig. 5c).

**Discussion**

We note that the optical mirage demonstrated here is different from imaging[16]. For imaging, its physical essence is to make a limited number of point sources at the input plane to focus at the output plane. Once designed well, the imaging lens can be naturally applied to an arbitrary object with a certain resolution. For optical mirage, it is almost impossible to be universal like that, because optical mirage is site-specific for a given input/output sequence or some input/output categories. For different scenarios and users, the input and output may be different, for

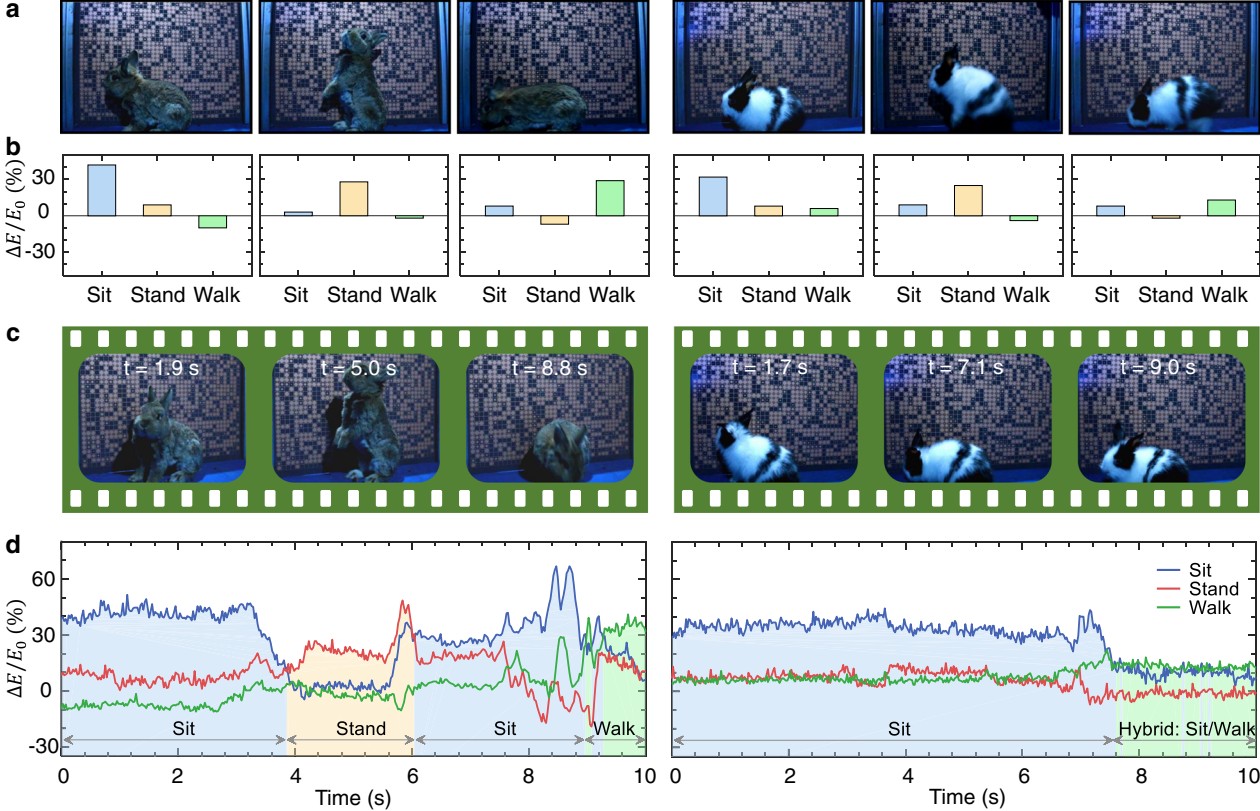

**Fig. 3 Experimental results showing real-time recognition of the postures of two rabbits. a** Snapshots of two rabbits having different postures and sizes, corresponding to the average experimental results in **b**. We quantify the accuracy of the results using the variance of the electric field amplitude over its original amplitude, that is, $\Delta E/E_0$. **c** Snapshots of two freely-playing rabbits; see the full videos in Supplementary Movies 1 and 2. **d** Experimental time-varying results. For the gray rabbit (left), the signal varies dramatically, where the three postures are clearly observed. For the smaller black-white rabbit (right), the signal remains relatively stable and shows hybrid postures toward the end.

example, converting a moving rabbit into a moving giraffe, cat, and tiger. In real-world applications, we may reasonably incorporate more images into large-scale neuro-metamaterials as an optimization task (Supplementary Figs. 5 and 6), or treat optical mirage as an inference task for untrained input (Supplementary Fig. 7). The result in Supplementary Fig. 6 shows that the optical illusion can still work for a running giraffe (for the case where the similarities among the output images are weak). For a certain network, when the number of input images increases, the SSIM of output images may decrease. Taking the small-scale network and an experimentally recorded movie as examples, the length of the input sequence reaches about 150 images with an average SSIM of 75%.

In conclusion, we proposed and experimentally realized a more ambitious 3D object-recognition strategy enabled by neuro-metamaterials, i.e., passive and inanimate metamaterials endowed with cognitive and computing ability. In contrast to conventional object-recognition systems, ours is realized entirely in optics and for 3D targets; the additive back-end detection system is used only to illustrate the perception result for users and is not a part of the recognition calculations. In other words, our approach successfully integrates two functionalities: the capture and processing of optical images. In the experiment, we utilized layered neuro-metamaterials for identifying the postures of two freely-playing rabbits without human intervention, which can strongly verify the preliminary metrics of the neuro-metamaterials. Furthermore, we introduced an intriguing optical mirage mechanism, converting a sequence of rabbit movements into different dynamic holography. This novel scheme substantially simplifies the mainstream yet difficult-to-reach transformation

optics-based optical illusion, because of its practical feasibility and dynamic input.

Our work provides deep insight into and a better understanding of the way metamaterials can facilitate real-world applications and may trigger other existing or yet-to-be conceived applications, in terms of wireless communication[22], illusive cloak[23,39,40], and information processing and display[24,25]. In some applications that involve high-throughput and on-the-fly data processing, low-energy consumption, and low-heat generation, neuro-metamaterials may find definite advantages over electronic counterparts, such as accelerating matrix calculation in artificial neural network[15], passive imaging pre-processor[11–13], and synthetic aperture radar[41]. Looking forward, we can also envision that the neuro-metamaterials may provide a large degree of freedom to generate a highly-informative holography technology in tandem with broadband metasurfaces[42], orbital angular momentum[25], reconfigurable technologies[43], and nonlinear components[44]. In addition to spatially separated metasurfaces, compact metamaterials and nanophotonics[45] may also offer a fertile platform to foster integrable, compact, and speed-of-light neuro-metamaterials, ushering in a possible "Optical Internet of Things" era.

## Methods

**Data collection**. We considered four parameters of a rabbit to generate the simulated data, i.e., the rabbit's size, posture, rotation angle, and distance to the neuro-metamaterials; see Supplementary Fig. 1. We import these models into the commercial software package CST Microwave Studio and continuously generate the data using the MATLAB-CST co-simulation method. The simulated data are shuffled and 80% are blindly selected as the training set and the remaining 20% are used for testing the performance of the neuro-metamaterials.

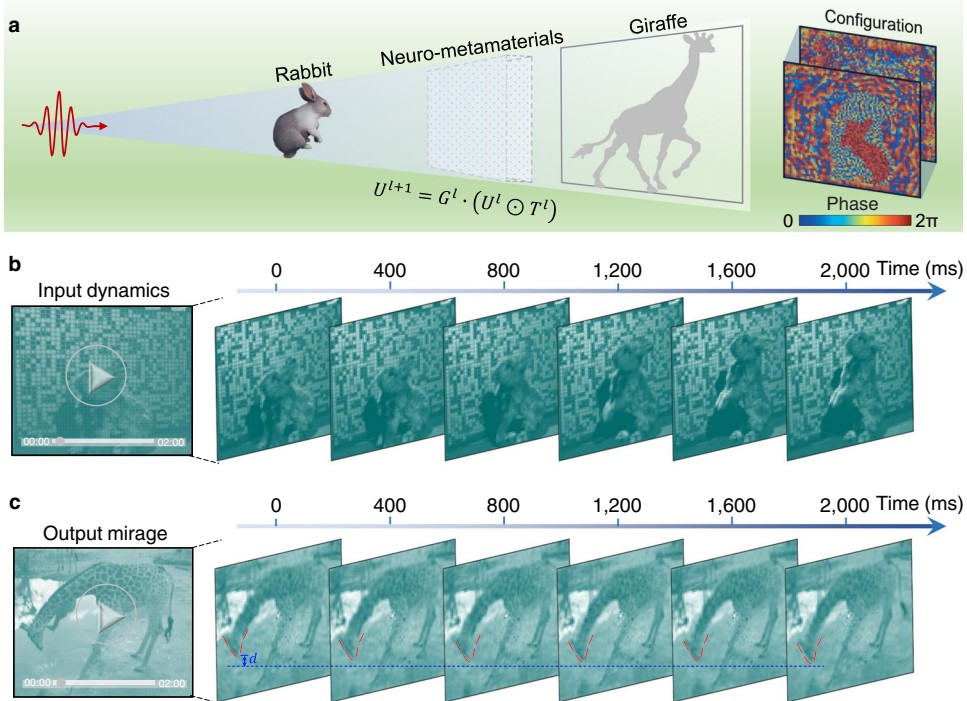

**Fig. 4 Dynamic optical mirage. a** Schematic of a novel optical mirage enabled by neuro- metamaterials. As compared with the mainstream transformation optics-based optical illusion, such a scheme is more practically feasible and valid for both static and non-static objects. In this example, we aimed to convert a sequence of rabbit movements into a mirage of a giraffe. The final training results are shown in inset on the right-hand side of **a**. The formula for the inset generalizes the diffractive procedure; see details in Supplementary Note 1. **b** Snapshots from the video used as the input of the neuro-metamaterials (Supplementary Movie 1). **c**, Output mirage, implanting the rabbit's motion on a giraffe drinking alongside a river. To clearly show this, we added a horizontal line in **c**, where the distance between the giraffe's mouth and the horizontal line is labeled as $d$. It is obvious that $d$ becomes smaller and smaller, and ultimately reaches zero. The entire dynamic of this mirage and the raw images can also be vividly observed in Supplementary Movie 3.

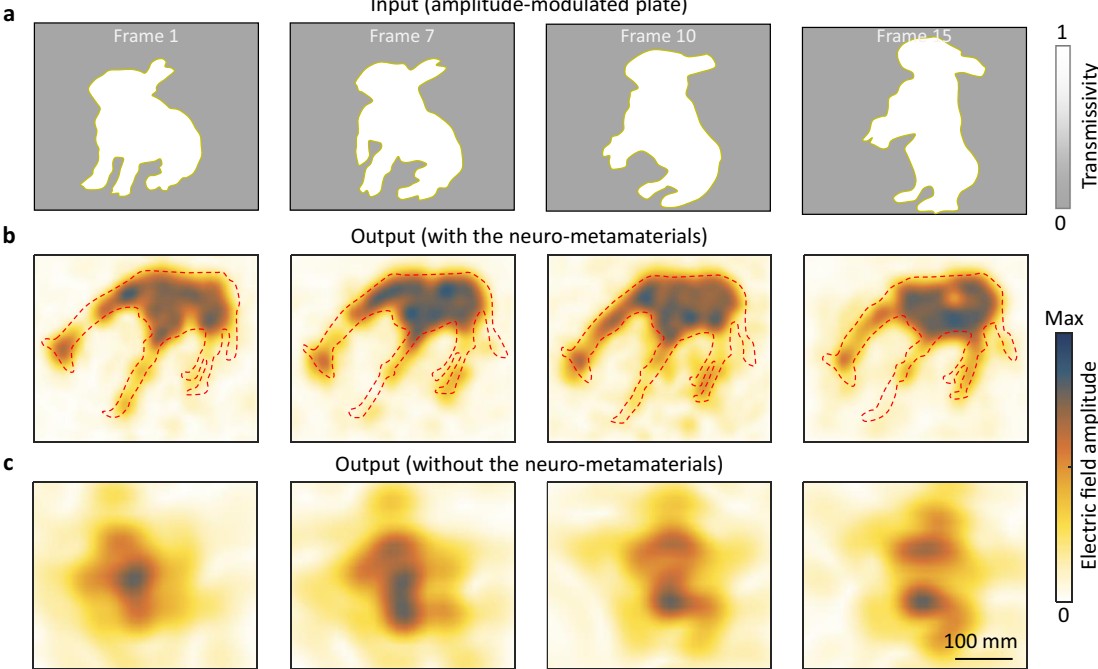

**Fig. 5 Experimental demonstration of dynamic optical mirage. a** Input rabbit's plates of frames 1, 7, 10, and 15. Input plates encode the amplitude of the input field. **b** Measured electric field amplitude with the neuro-metamaterials at the output plane. The results clearly show a mirage of giraffe, and the giraffe's head goes down gradually. **c** Measured electric field amplitude without the neuro-metamaterials.

**Training of neuro-metamaterials**. The neuro-metamaterials are trained using Python version 3.5.0. and TensorFlow framework version 1.10.0 (Google Inc.) on a server (GeForce GTX TITAN X GPU and Intel(R) Xeon(R) CPU X5570 @2.93 GHz with 48GB RAM, running a Linux operating system). It takes dozens of minutes for our neuro-metamaterials to converge. In the future, it will be possible to embed nonlinear components in the neuro-metamaterials to extend their capability[44]. A similar idea can be applied to optical neuromorphic computing[45].

**Cross-validation**. To validate that our network is reliable, we use a cross-validation method. To be specific, we divide the whole dataset into five parts. Each time, we take one of them as the test set (20%) and the rest as the training set (80%). This way, we obtain five sets of accuracy rates in total. We then calculate the average accuracy rates of training (98.8%) and test sets (97.6%), which are close to those in the main text. This provides another ground to validate our clarification.

**Experimental measurement setup**. A high-directivity lens antenna centered at the neuro- metamaterials was used as the excitation source. Three monopole probes, which we built ourselves, were connected to one RF switch (HMC641ALC4), collecting the amplitude signals from the three ports at microseconds[37]. The received signal was amplified by a broadband amplifier and down-converted by 6 GHz. Then, we employed an AD9361 as the RF processor, containing a low noise amplifier, mixer, and other components, and used a Xilinx ZYNQ for data processing accelerated by a field-programmable gate array. These electronic devices are used only to illustrate the perception results in microwave and are not a part of the object-recognition process.

## Data availability

Data presented in this publication is available on Figshare with the following identifier (https://doi.org/10.6084/m9.figshare.19602145.v1).

## Code availability

The codes used in the current study are available from the corresponding authors upon reasonable request.

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

## Acknowledgements

We thank A. L. Wang for fruitful discussion. The work at Zhejiang University was sponsored by the National Natural Science Foundation of China (NNSFC) under Grants Nos. 62101485 (C.Q.), 61625502 (H.C.), 11961141010 (H.C.), 61975176 (H.C.), 62071423 (B.Z.), 62071424 (E.L.), and 62027805 (E.L.), the Top-Notch Young Talents Program of China (H.C.), and the Fundamental Research Funds for the Central Universities (H.C.).

## Author contributions

C.Q. conceived the idea and conducted numerical simulations and experiments. Z.W. helped set up the experiment. C.Q., I.K., H.Q., and H.C. contributed extensively to the

writing of the manuscript. I.K., H.Q., T.C., B.Z., X.L., Y.S., E.L., and H.C. analyzed data and interpreted the details of the results. C. Q., E.L., and H.C. supervised the project.

## Competing interests

The authors declare no competing interests.
