## [Peer Review File · Nature Communications]

Dynamic recognition and mirage using neuro-metamaterialsREVIEWER COMMENTS

Reviewer #2 (Remarks to the Author):

[Overview]

This manuscript demonstrates dynamic object recognition and optical mirage using metamaterial layers. Specifically, recognition of motions of a rabbit (sit, stand, walk) was demonstrated using multi-layered metamaterial, whose structures were inversely designed using deep learning. This work shows interesting applications of multi-layered metamaterials, which would be attractive for a broad audience.

[Comment 1]

Personally, it is a bit difficult to imagine how the metamaterial looks like. In the manuscript, the specifications of metamaterials were given as:

- each layer consisted of 30 x 40 neurons
- physical dimension of each neuron was 13 mm x 13 mm x 3 mm
- working frequency was 8.6 GHz
- distance between each adjacent layer was 300 mm
- the number of layers was "L", not given as a specific number

From the given specs, the physical dimension of each layer would be $(30 \times 13 \text{ mm}) \times (40 \times 13 \text{ mm}) = (390 \text{ mm}) \times (520 \text{ mm})$, but the distance between each layer seems a bit large compared to the size of each layer. From the schematics of the metamaterial (Fig. 2a), the distance may be incorrectly given. Also, the number of layer could be given as a specific number.

[Comment 2]

In line 62, the authors mention "Moore's law" regarding the limited speed of electronics. However, as far as I know, Moore's law concerns the number of transistors in a chip. Also, the authors may need to elaborate more in detail how electronics becomes the bottleneck of "the speed of the entire system". Is this because of the clock rate of electronics, which is limited to 3~5 GHz? Or, is this because of the nature optical processing units, where the computation is performed by scattering of light passing through the metamaterial?

[Comment 3]

The authors emphasize the superior speed of optical circuitry for real-time imaging processing applications. However, imaging processing and recognition techniques nowadays are quite advanced and sophisticated, such that they can be used for self-driving, facial recognition, and so on. This manuscript provides proof-of-concepts where image recognition is realized using optical metamaterials, but the demonstrated applications can also be realized using electronics. The authors may provide some potential specific applications where electronics cannot work, or optical processing has definite advantages.

Reviewer #3 (Remarks to the Author):

In this work the authors proposed and experimental demonstrated a neuro-metamaterial based on an optimization of phase distribution on a metasurface. The performance of the proposed neuro-metamaterial is verified experimentally via realtime detection of posture of a rabbit (one-hot coding, three sensors). However, some important problems must be addressed before I could give the paper a further consideration:

1. In the section of “Prospective application scenario and physical mechanism of neuro-metamaterials” the authors give an outlook of the potential application of the proposed method in security check (see Fig. 1). However, they should instead consider the feasibility of applying their strategy in a more realistic scenario such as the detection of concealed weapons that cannot be seen directly.

2. The concept of optical illusion is interesting which exemplifies here a realtime projection of a giraffe image as input by a rabbit image. An experimental demonstration of such an effect would definitely more convincing.

3. In both the Abstract and Introduction sections the authors state they have inversely designed the metasurface. This is a bit confusing. Do they really inversely designed the unit cell structure? Or only the desired effective phase distribution the metasurface needs to produce? If they just obtain the phase distribution of the neuro-metamaterial, I think it is better to use ‘optimize’ instead of ‘inverse design’. Please clarify it.

4. In line 125 on page 5 the authors state the size of unit cell is 13 mm and the working frequency is 8.6 GHz which corresponds to a wavelength of 34 mm. In other words, the unit cell is as small as $1/3$ wavelength approximately. But when discussing optical illusion (page 7, line 189) they only provide the number of pixels, without any explanation on the pixel size and incident wavelength. If the values of these two parameters are equal to what they used on page 5, they have realized a super-resolution imaging which, however, are supposed to be unattainable based on their mechanism.

5. On page 5, line 144: they state “Once trained, the operation of the neural network is universal for both microwave and visible frequencies, provided that suitably designed metamaterials are used.” But I have not been convinced that their trained neuro-metamaterial is a broadband device. It seems that they have neither introduced the parameter of frequency into their wave equations which only

correspond to a single frequency nor demonstrated any broadband functionality in the following experiments.

6. How many frames are there in the video showing a giraffe drinking water? In line 187 (page 7) the authors state the input video contains 15 pictures, but giraffe is only related to 2 out of them. I therefore have a significant concern on the fundamental limitations in such a dynamic optical illusion, and suspect whether or not it can only be realized for very limited types of images (considering the capability of such kind of neural network).

7. In Fig. S3 the authors show the transmitted spectra. Is this suggesting a broadband functionality? If the answer is yes, I think it is necessary to clarify how to achieve a broad bandwidth.

Response Letter to Reviewers

We are grateful for the constructive comments on this manuscript (NCOMMS-21-15025-T) from all the referees. In the text below, each comment is quoted in *italics* and is followed by the corresponding detailed response. We have also revised the manuscript and supplementary material accordingly. These updates are highlighted in blue and by a vertical red line in the left margin in those files. In the text below, the references to these updates are highlighted in a similar way (i.e., by a vertical red line).

General comments from Referee #2:

[Overview] This manuscript demonstrates dynamic object recognition and optical mirage using metamaterial layers. Specifically, recognition of motions of a rabbit (sit, stand, walk) was demonstrated using multi-layered metamaterial, whose structures were inversely designed using deep learning. This work shows interesting applications of multi-layered metamaterials, which would be attractive for a broad audience.

Authors Response:

We thank the referee for his/her positive comments, and acknowledging “[this work] would be attractive for a broad audience”. In the following, we address the specific comments point-by-point whilst revising our manuscript.

Specific comments from Referee #2:

Referee #2 -- Comment 1:

[Comment 1] Personally, it is a bit difficult to imagine how the metamaterial looks like. In the manuscript, the specifications of metamaterials were given as:

- each layer consisted of 30 x 40 neurons
- physical dimension of each neuron was 13 mm x 13 mm x 3 mm
- working frequency was 8.6 GHz
- distance between each adjacent layer was 300 mm
- the number of layers was "L", not given as a specific number

From the given specs, the physical dimension of each layer would be (30x13 mm) x (40x13 mm) = (390 mm) x (520 mm), but the distance between each layer seems a bit large compared to the size of each layer. From the schematics of the metamaterial (Fig. 2a), the distance may be incorrectly given. Also, the number of layer could be given as a specific number.

Authors Response:

We appreciate the referee for his/her careful reading. The figure we show is a general schematic of the neuro-metamaterials, which is composed of layered deliberately-designed metasurfaces. To make it consistent with the recognition setup, we have followed the referee’s suggestion, and replaced Fig. 2a with Fig. R1 by enlarging the distance between layers and labelling the layer number (two hidden layers, i.e., $L = 2$).

Fig. R1 | Schematic of neuro-metamaterials for recognition of a rabbit's postures. In our case, two hidden layers are designed, each of which is composed of stacked metasurfaces.

Referee #2 -- Comment 2:

[Comment 2] In line 62, the authors mention "Moore's law" regarding the limited speed of electronics. However, as far as I know, Moore's law concerns the number of transistors in a chip. Also, the authors may need to elaborate more in detail how electronics becomes the bottleneck of "the speed of the entire system". Is this because of the clock rate of electronics, which is limited to 3~5 GHz? Or, is this because of the nature optical processing units, where the computation is performed by scattering of light passing through the metamaterial?

Authors Response:

Thanks for the good questions. We agree that "Moore's law concerns the number of transistors in a chip", but note that Moore's law also regards to the computing speed of electronics. For example, "this [Moore] law states that processor speeds, or overall processing power for computers will double every two years" [Nature 530, 144–147 (2016)] & "Moore's law states that the number of transistors on a microprocessor chip will double every two years or so —which has generally meant that the chip's performance will, too" [http://www.moorelaw.org/]. In the following, we explain this point in detail and answer the question "how electronics becomes the bottleneck...".

Electronic computing is implemented by switching on/off millions of transistors in electronic processor. The processing speed is mainly affected by integration and energy consumption (in general, they are manifested as clock rate). A high-integration electronic processor typically enables a high processing speed, because the miniaturization of feature size makes the parasitic capacitance smaller and thus a smaller RC delay. However, the twilight of Moore's law makes the integration difficult to increase. Another important point that affects the electronic computing speed is energy consumption. Due to the induced heat issue, the clock rate—how fast microprocessors execute instructions—is typically limited to GHz scale [Nature 530, 144–147 (2016)].

Optical computing is performed by light scattering (at the speed of light); after the light passes through a well-designed metamaterial, the computing result can be obtained immediately without the switching delay. For the application of object recognition, conventional approaches (completely or partially) rely on electronic computing. For every approach, the entire computing speed will be limited by an electronic processor [Nat. Photon. 4, 261–263 (2010)]. This is why we clarified in the main text that "the speed of the entire system is ultimately limited by the speed of the electronic input/output".

To avoid possible misunderstanding, in the new submission, we have made following updates.

On lines 61-63, “The speed of the entire recognition system is ultimately limited by the frequency of an electronic processor [4]. The processing speed is difficult to increase due to the energy consumption of the processor, becoming ever more challenging at the twilight of Moore’s law [19].”

Referee #2 -- Comment 3:

[Comment 3] The authors emphasize the superior speed of optical circuitry for real-time imaging processing applications. However, imaging processing and recognition techniques nowadays are quite advanced and sophisticated, such that they can be used for self-driving, facial recognition, and so on. This manuscript provides proof-of-concepts where image recognition is realized using optical metamaterials, but the demonstrated applications can also be realized using electronics. The authors may provide some potential specific applications where electronics cannot work, or optical processing has definite advantages.

Authors Response:

Indeed, electronic computing is quite advanced and sophisticated nowadays. However, the energy consumption and physical limitation of transistor hinder its further development. In this context, scientists are looking for new computing modality that may alleviate these problems. Optical computing has found to be a promising candidate due to its unique features of speed-of-light operation, low-power consumption, and parallel capability. Given the strengths of optical computing, its main applications are not in replacing electronic computers but in narrower niches in which optical advantages are the greatest. A friendly relationship between them should be more complementary than competitive [*Nat. Photon.* 4, 261–263 (2010)].

Optical computing may find definite advantages in some applications involving high-throughput and on-the-fly data processing, low-energy consumption, and low-heat generation. Here, we take three specific examples. **First**, accelerating matrix calculation in artificial neural network. Matrix calculation is one of the fundamental mathematical operations in machine learning. Conventional matrix calculation is completed by an electrical processor, with limited speed and high energy consumption. Optical computing can easily encode data in parallel to the input light and obtain calculation result at light speed, leading to a great improvement of computing speed and reduction of energy loss [*Nat. Photon.* 11, 441–446 (2017)]. **Second**, passive imaging pre-processor. Many imaging applications require frequent use of spatial differentiation, integration, and convolution [*Science* 343, 160-163 (2014)]. For example, using spatial differentiation to extract imaging edge and geometric features. However, in many applications that require on-the-fly imaging processing, such as in autonomous driving, medical and satellite applications, essential high-throughput and power-saving edge detection remains a key challenge. Passive optical neuro-metamaterials may be an attractive choose [*Nat. Commun.* 8, 15391 (2017) & *PNAS* 116, 11137-11140 (2019)]. **Third**, synthetic aperture radar (SAR). SAR systems typically generate copious amounts of complex raw data difficult to compress and require intense computation to produce high-resolution image. Optical processor provides inherent parallel computing capabilities that could be used advantageously for, such as SAR spatial filter and Fourier transform. This could provide benefits for real-time decisions in navigation strategy, automatic instruments orientation, and moreover, for interplanetary missions or unmanned aerial vehicles [*Proc. SPIE* 8714, 871416 (2013)].

In the new submission, we have added some sentences.

On lines 226-229, “In some applications that involve high-throughput and on-the-fly data processing, low-energy consumption, and low-heat generation, neuro-metamaterials may find definite advantages over electronic counterpart, such as accelerating matrix calculation in artificial neural network [15], passive imaging pre-processor [11-13], and synthetic aperture radar (SAR) [32].”

General comments from Referee #3:

In this work the authors proposed and experimental demonstrated a neuro-metamaterial based on an optimization of phase distribution on a metasurface. The performance of the proposed neuro-metamaterial is verified experimentally via realtime detection of posture of a rabbit (one-hot coding, three sensors). However, some important problems must be addressed before I could give the paper a further consideration:

Authors Response:

We are grateful to the referee for the detailed review and constructive comments that we answer point-by-point below.

Specific comments from Referee #3:

Referee #3 -- Comment 1:

1. In the section of "Prospective application scenario and physical mechanism of neuro-metamaterials" the authors give an outlook of the potential application of the proposed method in security check (see Fig. 1). However, they should instead consider the feasibility of applying their strategy in a more realistic scenario such as the detection of concealed weapons that cannot be seen directly.

Authors Response:

Thanks for the good suggestion. We have re-plotted Fig. 1 (Fig. R2) to have a concealed gun, and discussed it on lines 95-97, "Since microwaves can readily penetrate most clothing material, a man carrying prohibited items (such as concealed weapons) will induce strong and distinct EM scattered waves... [24]."

Fig. R2 | Illustrative usage of neuro-metamaterials in a security inspection system.

Referee #3 -- Comment 2:

2. The concept of optical illusion is interesting which exemplifies here a realtime projection of a giraffe image as input by a rabbit image. An experimental demonstration of such an effect would definitely more convincing.

Authors Response:

We are glad that the referee considers "the concept of optical illusion is interesting". Following the referee's suggestion, we have added a new experiment to demonstrate optical illusion. As shown in Fig. R4a, we first

picked four distinct frames from the original input video, and extracted the rabbit's outline via edge detection. Then, we resized the frames into 325 mm × 360 mm (i.e., 25*30 unit cells) plates as the input of neuro-metamaterials. Input plates encode the amplitude of the input field. Finally, we fabricated the pre-trained neuro-metamaterials and measured the electric field distributions at the output plane; see the experimental setup in Fig. R3. The measured results in Fig. R4b successfully converts a sequence of rabbit movements into a mirage of a giraffe, and the giraffe's head goes down gradually. For comparison, we also measured the electric field distributions without the neuro-metamaterials, which shows a completely different result (Fig. R4c).

In the new version, we have added Fig. R4 as Fig.5 and above description on lines 204-210.

Fig. R3 | Experimental setup of optical illusion. A high-directivity lens antenna excites a plane wave toward the neuro-metamaterials with a cross section of 325 mm × 360 mm. Behind the neuro-metamaterials, a small monopole probe moves automatically to scan the spatial intensity distribution at the output (xoz) plane.

Fig. R4 | Experimental results of optical illusion. a, Input rabbit's pattern. b, Output with the neuro-metamaterials, where the giraffe's head goes down gradually. c, Output without the neuro-metamaterials.

Referee #3 -- Comment 3:

3. In both the Abstract and Introduction sections the authors state they have inversely designed the metasurface. This is a bit confusing. Do they really inversely designed the unit cell structure? Or only the desired effective phase distribution the metasurface needs to produce? If they just obtain the phase distribution of the neuro-metamaterial, I think it is better to use 'optimize' instead of 'inverse design'. Please clarify it.

Authors Response:

We design the phase distribution of the neuro-metamaterials. Following the referee's suggestion, we have rephrased the expression 'inverse design' into 'optimize' in the new submission.

Referee #3 -- Comment 4:

4. In line 125 on page 5 the authors state the size of unit cell is 13 mm and the working frequency is 8.6 GHz which corresponds to a wavelength of 34 mm. In other words, the unit cell is as small as 1/3 wavelength approximately. But when discussing optical illusion (page 7, line 189) they only provide the number of pixels, without any explanation on the pixel size and incident wavelength. If the values of these two parameters are equal to what they used on page 5, they have realized a super-resolution imaging which, however, are supposed to be unattainable based on their mechanism.

Authors Response:

We would like to note that **we have not realized super-resolution imaging** in this system. The reason could be understood from Fig. R5. For each element in the phase mask, the numerical aperture (NA, i.e., $n \cdot \sin\theta$) is always smaller than one, so that high-k components are impossible to exist.

Regarding the pixel size and incident wavelength in optical illusion, they are equal to those on page 5. Hence, the pixel size is $\sim 1/3\lambda$. However, we want to explain that **this pixel size actually refers to the spatial sampling period** on the output plane. We can freely increase/decrease the spatial sampling period, but it does not relate to super-resolution imaging. To avoid possible misunderstanding, we have made following revisions.

On lines 193-195, "the raw input and output images were spatially sampled by 70×84 points, and the distance between points is set to be 13 mm (the size of the neuron)."

On line 198, "the incident wavelength was set to be the same as that in Fig. 2."

Fig. R5 | Numerical aperture of the phase mask. n is the refractive index of the medium between the objective lens and the object (in our case, $n = 1$ for air) and θ_i is half the angular aperture.

Referee #3 -- Comment 5:

5. On page 5, line 144: they state "Once trained, the operation of the neural network is universal for both microwave and visible frequencies, provided that suitably designed metamaterials are used." But I have not

been convinced that their trained neuro-metamaterial is a broadband device. It seems that they have neither introduced the parameter of frequency into their wave equations which only correspond to a single frequency nor demonstrated any broadband functionality in the following experiments.

Authors Response:

We apologize for the possible confusion this sentence may cause. The point we want to convey is that, once trained, the resulting neural network can be scaled to fit to any frequency in theory. In experiment, we need to design specific metasurfaces to construct the pre-trained neural network, which typically works in a narrow band. Below, we elaborate on the design principle to better explain this point.

As schematically shown in Fig. R6a, we take a two-layer neural network (the space separation is b) as an example. The number of unit cell is $m \times n$, and the size of unit cell is $a \times a$. Our goal is to optimize the transmitted phase ($\phi_{m,n}$) of each unit cell. In this process, each pixel can be regarded as a secondary source (following Rayleigh-Sommerfeld diffraction), and the electric field of the pixel located at $\vec{r}^{l+1}(x, y, z)$ is contributed by all the pixels of the low-level layer (see Supplementary Note 1 for details),

$$u(\vec{r}^{l+1}) = \sum_{m=1}^M \sum_{n=1}^N \frac{-1}{2\pi} (ikR - 1) e^{ikR} \cdot u_{m,n}^l \cdot t_{m,n}^l \cdot \frac{b \cdot a \cdot a}{R^3} \quad (S1)$$

where R is the distance between the observation location \vec{r}^{l+1} and the source location \vec{r}_s^l (m -th row, n -th column, l -th layer). $t_{m,n}^l$ represents the transmitted spectrum of the source location \vec{r}_s^l . $k = 2\pi/\lambda$ is the wave vector of light in free space. If the wavelength λ changes, $u(\vec{r}^{l+1})$ remains consistent as long as R (a and b) changes correspondingly. Thus, the neural network (phase mask) is scalable to any frequency.

In experiment, we need to design specific metasurfaces to satisfy the transmitted phase of each unit cell. However, the designed metasurfaces (rely on electronic/magnetic resonance) typically work in a narrow band around wavelength λ_1 (Fig. R6b). To avoid possible misunderstanding, we have rephrased the sentence in the new submission, on lines 147-149, "Once trained, the resulting neuro-metamaterials can be scaled to fit to any frequency in theory. In experiment, one needs to design specific neuro-metamaterials to construct the pre-trained neural network, which typically works in a narrow band."

Fig. R6| Design principle of the neuro-metamaterials. a, Layout of the phase mask, which is scalable to any frequency. **b**, Layout of the neuro-metamaterials for recognition, working at a single wavelength λ_1 .

Referee #3 -- Comment 6:

6. How many frames are there in the video showing a giraffe drinking water? In line 187 (page 7) the authors state the input video contains 15 pictures, but giraffe is only related to 2 out of them. I therefore have a significant concern on the fundamental limitations in such a dynamic optical illusion, and suspect whether or not it can only be realized for very limited types of images (considering the capability of such kind of neural network).

Authors Response:

The output video contains 15 frames, which one-to-one correspond to the input frames. To clearly show this, we added a horizontal line in Fig. 4 (Fig. R7), where the distance between the giraffe's mouth and the horizontal line is labelled as d . It is obvious that d becomes smaller and smaller, and ultimately reaches zero (rather than "only related to 2 out of them"). This continues changes can also be vividly observed in Supplementary Video S3. We guess the unclear marks in original Fig. 4 may confuse the referee. In the new version, we have updated Fig. 4 and made revisions as following,

On line 192, "an online giraffe video (also contains 15 frames)"

On line 200, "with its head going down gradually."

In the caption of Fig. 4, "To clearly show this, we added a horizontal line in c, where the distance between the giraffe's mouth and the horizontal line is labelled as d . It is obvious that d becomes smaller and smaller, and ultimately reaches zero."

Fig. R7| Dynamic optical mirage. a, Snapshots from the video used as the input of the neuro-metamaterials. b, Output mirage, implanting the rabbit's motion on a giraffe drinking alongside a river.

Referee #3 -- Comment 7:

7. In Fig. S3 the authors show the transmitted spectra. Is this suggesting a broadband functionality? If the answer is yes, I think it is necessary to clarify how to achieve a broad bandwidth.

Authors Response:

Thanks for the careful reading and useful comment. The transmitted spectra in Fig. S3 do not suggest a broadband functionality. As explained in **Referee #3 -- Comment 5**, the meaning of broadband refers to the possibility to scale the design to fit it for other wavelengths. In experiment, we always need to design specific metamaterials to satisfy the desired transmitted spectra, each typically working in a narrow band. In the new submission, we have clarified this point in Supplementary Note 3.

“These phase profiles can be scaled to fit to any frequency in theory. In experiment, we need to design specific neuro-metamaterials to construct the pre-trained neural network, which typically works in a narrow band.”

REVIEWER COMMENTS

Reviewer #2 (Remarks to the Author):

The manuscript entitled “Dynamic recognition and mirage using neuro-metamaterials” proposes all-optical rabbit posture classification metasurfaces system that operates in the microwave domain and dynamic optical mirage system which converts rabbit movements to a mirage of giraffe. Although constructing such diffractive neural network system has already been demonstrated with different tasks [Science 361, 1004, (2018), LSA 8, 112 (2019), PRL 123, 023901 (2019), LSA 10, 14 (2021), Nat. Photon. 15, 367 (2021), arXiv:2107.07873], there are three main novelties of this work. First, these neuro-metamaterials can process images of 3D objects. Second, the optical platform was well-demonstrated in the actual experiment with high accuracy. Finally, dynamic optical mirage to convert a sequence of rabbit movements to a mirage of a giraffe showed a successful performance. Overall, the revision is well done, but some important remarks are to be issued especially in parts related to application examples. The followings are suggested.

- I think the suggested prospective applications are insufficient to appeal the advantage of neuro-metamaterial system. For example, the authors suggested an idea of replacing the scanning systems in the public transport security. However, such situations do not require a light-speed processing or computationally extensive image processing. Current electronic systems are working well with no considerable problems. Other appropriate application examples are required.
- In Figure S1, the notation in the caption does not match with the corresponding figure. While there are no explicit graphical descriptions about (x_m, y_l, z_n) and (x_r, y_{l+1}, z_s) in the figure, it is written in the captions. This is a bit confusing. Please be consistent with superscripts and subscripts in the figure, figure caption and the manuscript.
- On line 61, "clock speed" or "clock rate" may be better than a less specific term "frequency".
- On line 117, I recommend explaining in more detail how human and rabbit resemble biologically.
- On line 135, this work mentions “the trained neural network is reliable with little overfitting”. However, more grounds are required to claim this network has little overfitting.
- On line 144, it may be more clear to specify that two layers of metamaterials were used ($L = 2$).

- On lines 147-149, the frequency change by using scale change is only applicable when the material property is dispersionless. To the best of our knowledge, material properties in microwave regime is quite different from visible and infrared frequencies, so the scale change cannot be applied as easily.

- On line 219, it is mentioned that the experiment achieves 98% accuracy utilizing layered neuro-metamaterials for identifying the rabbits' postures. It should be clearly explained how the quantitative accuracy was obtained in the experiment. I assume the evaluation was done by recording the video of the rabbits and comparing with its prediction signal. However, if that's the case, it could have been evaluated only on certain, selected frames, which is not appropriate. Thus, the method of calculating the experimental accuracy should be specified.

- I wonder if these are typos:

In line 227, "Geforce '249 10' GTX"

In line 228, "a Linux '250' operating"

'249 10' and '250' are also written in the supplementary.

Reviewer #3 (Remarks to the Author):

Thanks for the revisions made by the authors in the last round in light of the referees' reports. However, my major concern is in the novelty of this work over the existing ones, which have not yet been well addressed in the revised manuscript. Thus, unfortunately, I cannot recommend its publication in Nature Communications in its current form. The main issues are identified below:

1. In the authors' reply to my comments 5 and 7, their strategy applies to EM waves of different frequencies, while the metasurfaces designed based on it only works for a single frequency, which makes their statements of broadband functionality less convincing. Actually, their mechanism shows high similarity to the ones found in the literatures (e.g., Science 361, 1004–1008 (2018), Photon. Res. 7, 823-827 (2019) and Nat. Comm. 11, 6309 (2020) which respectively proposed the architecture of optical neuro-metamaterial described here, the optical artificial neural inference with dynamic effects and the metasurface-based passive neural network with functionality of recognizing object's shape).

2. In their reply to my comment 6, the method for producing dynamic illusion is innately identical to the imaging effect demonstrated in the previous work (see the Supplemental Materials of Science 361,

1004–1008 (2018)) which refers to a correspondence between the input and output images and imaging quality of illusion. What is more, in the current work the input and output data only include 15 images, which is far less than what was demonstrated there. Obviously, here the generalization ability of a neural network cannot be verified when the training and testing databases are same.

3. The authors have not clarified the questions about the fundamental limitations on the optical dynamic illusions I raised in the last round. Please demonstrate the relationship between the frame rate and accuracy of illusion based on the current framework.

Response Letter to Reviewers

We are grateful for the constructive comments on this manuscript (NCOMMS-21-15025A) from all the referees. In the text below, each comment is quoted in *italics* and is followed by the corresponding detailed response. We have also revised the manuscript and supplementary material accordingly. These updates are highlighted in blue and by a vertical red line in the left margin in those files. In the text below, the references to these updates are highlighted in a similar way (i.e., by a vertical red line).

General comments from Referee #2:

The manuscript entitled “Dynamic recognition and mirage using neuro-metamaterials” proposes all-optical rabbit posture classification metasurfaces system that operates in the microwave domain and dynamic optical mirage system which converts rabbit movements to a mirage of giraffe. Although constructing such diffractive neural network system has already been demonstrated with different tasks [Science 361, 1004, (2018), LSA 8, 112 (2019), PRL 123, 023901 (2019), LSA 10, 14 (2021), Nat. Photon. 15, 367 (2021), arXiv:2107.07873], there are three main novelties of this work. First, these neuro-metamaterials can process images of 3D objects. Second, the optical platform was well-demonstrated in the actual experiment with high accuracy. Finally, dynamic optical mirage to convert a sequence of rabbit movements to a mirage of a giraffe showed a successful performance. Overall, the revision is well done, but some important remarks are to be issued especially in parts related to application examples. The followings are suggested.

Authors Response:

We thank the referee for his/her positive comments and clearly pointing out the main novelties of our work. The mentioned references have been appropriately cited in the new version (e.g., Refs. [16,17,18,35,S2,S3]). In the following, we address the specific comments point-by-point whilst revising our manuscript.

Specific comments from Referee #2:

Referee #2 -- Comment 1:

- I think the suggested prospective applications are insufficient to appeal the advantage of neuro-metamaterial system. For example, the authors suggested an idea of replacing the scanning systems in the public transport security. However, such situations do not require a light-speed processing or computationally extensive image processing. Current electronic systems are working well with no considerable problems. Other appropriate application examples are required.

Authors Response:

We understand the referee’s concern and appreciate him/her for the constructive suggestion. Current electronic systems indeed work well and they are mostly based on imaging technology. Imaging technology typically utilizes amplitude information of the scattered waves for data processing, in which some other information may be lost, like phase and polarization [Science 365, eaax1839 (2019)]. Our method starts from the very fundamental electromagnetic scattering waves to execute inference task at the speed of light. This procedure does not need the amplitude-to-imaging transform and is distinct from current imaging technology.

Therefore, for some applications (e.g., imaging polarimetry) or hard-to-discover object features (e.g., surface features, shape, shading, and roughness), neuro-metamaterials may find the advantages by directly and simultaneously utilizing multiply scattered wave properties, including phase, amplitude, and polarization, to reveal features that are otherwise invisible [Appl. Opt. 45, 5453–5469 (2006)]. We believe this method will greatly improve the performance of devices in dynamic recognition, for example, mitigating a long queue at airport security. Also, we envisage it serves as a complementary method to current electric security systems.

On the other hand, the example of security system is more about the illustration of the working principle and potential capability of neuro-metamaterials. The successful demonstration can also be transferred into other applications, particularly those involving speed-of-light operation, high-throughput data processing, and low-energy consumption. As suggested by the referee, here we take some specific examples. **First, intelligent prefilter in astronomy.** Astronomical radio observation and study on the galaxy formation typically necessitates vast data to be collected and processed [arXiv:2105.09943]. A representative achievement in this community is the WIDER supercomputer (<https://public.nrao.edu/gallery/the-widar-supercomputer/>). WIDAR uses FPGAs to perform correlations on the RF signal and only then sends relevant data to the cluster for further processing. However, electronic processing is limited by FPGA-setup times and electronic capacitive delay. To mitigate this, neuro-metamaterials may work synergistically with the WIDAR for intelligently sorting and correlating the signal looking for specific chunks of radio-patterns. In addition to the increased speed, neuro-metamaterials hold potential to directly exploit the wave-nature of the input RF signal to perform inherent correlation detection. **Second, driverless vehicles and autonomous drones.** Driverless vehicles and autonomous drones typically need to make split-second decisions to avoid collisions or deal with other unexpected events. However, this is challenging for many computers running complex AI systems. Besides, many computers need large amounts of onboard computing power and sensors, all of which add to weight and are a drain on scarce battery resources. Thus, to cater to the high demands on computing speed and power saving, neuro-metamaterials may play their role in imaging and classification.

In the new submission, we have incorporated this discussion on lines 113-118,

“Compared with electronic security systems (mostly based on amplitude imaging technology) [31], neuro-metamaterials may find the advantages by directly and simultaneously utilizing multiply scattered wave properties, including phase, amplitude, and polarization, to reveal features that are otherwise invisible [32]. Such neuro-metamaterials-enabled object recognition will be more convenient and immediate and can be readily extended to other applications, such as intelligent prefilter in astronomy and driverless vehicles and autonomous drones [33].”

Referee #2 -- Comment 2:

In Figure S1, the notation in the caption does not match with the corresponding figure. While there are no explicit graphical descriptions about (x_m, y_l, z_n) and (x_r, y_{l+1}, z_s) in the figure, it is written in the captions. This is a bit confusing. Please be consistent with superscripts and subscripts in the figure, figure caption and the manuscript.

Authors Response:

Thanks for the kind reminder. In the new version, we have updated the figure, figure caption and the manuscript to make them consistent. Please refer to Fig. R1 and the caption below.

Figure R1 | Mathematical model of the neuro-metamaterials. $\vec{r}_s^l = (x_s^l, y_s^l, z_s^l)$ represents the point (x_s^l, z_s^l) at the l -th layer. $\vec{r}^{l+1} = (x^{l+1}, y^{l+1}, z^{l+1})$ represents the point (x^{l+1}, z^{l+1}) at the $(l + 1)$ -th layer. The electromagnetic field of the meta-atom/neuron located at $(x^{l+1}, y^{l+1}, z^{l+1})$ is contributed by all the neurons of the low-level layer based on Huygens' principle.

Referee #2 -- Comment 3:

- On line 61, "clock speed" or "clock rate" may be better than a less specific term "frequency".

Authors Response:

Thanks for the good suggestion. We have rephrased the term "frequency" into "clock rate" on line 61.

Referee #2 -- Comment 4:

- On line 117, I recommend explaining in more detail how human and rabbit resemble biologically.

Authors Response:

Rabbit, as a member of the Lagomorpha order, is the closest phylogenetic relative to humans, next to primates. The high phylogenetic resemblance includes the body structure, electromagnetic responsiveness, and other congruences [J. Basic Clin. Physiol. Pharmacol. 29, 427–435 (2018)]. This is evident from numerous biomedical researches and optical imaging experiments. In microwave regime, electromagnetic wave can be reflected from the human/rabbit body and readily penetrate common clothing material/fur [IEEE Trans. Microwave. Theory Tech. 49, 1581–1592 (2001) & Sci. Rep. 11, 3545 (2021)]. Thus, we consider rabbit as research object for experimental convenience.

In the new submission, we have noted this point on lines 120-121.

"Because of its biological similarities (body structure, electromagnetic responsiveness, and other congruences) with human beings [34]."

Referee #2 -- Comment 5:

- On line 135, this work mentions "the trained neural network is reliable with little overfitting". However, more grounds are required to claim this network has little overfitting.

Authors Response:

A typical way to check whether the network has overfitting is by comparing the loss/accuracy between training and test sets. In this regard, our pre-trained neural network is reliable because the accuracy rates between the training (99%) and test sets (98%) are close. Following the referee's suggestion, we further validate this point using cross-validation method. To be specific, we divide the whole dataset into five parts. Each time, we take one of them as test set (20%) and the rest as training set (80%). This way, we obtain five sets of accuracy rates in total. We then calculate the average accuracy rates of training (98.8%) and test sets (97.6%), which are close to those in our main text. This provides another ground to validate our clarification.

In the new submission, we have added this discussion into Methods and noted it on lines 139-140,

"The close accuracy rates indicate that the neural network is reliable with little overfitting, which has also been checked by cross-validation (Methods)."

Referee #2 -- Comment 6:

- On line 144, it may be more clear to specify that two layers of metamaterials were used ($L = 2$).

Authors Response:

Following the referee's suggestion, we have specified the number of metamaterials on line 149.

"We considered two-layer neuro-metamaterials."

Referee #2 -- Comment 7:

- On lines 147-149, the frequency change by using scale change is only applicable when the material property is dispersionless. To the best of our knowledge, material properties in microwave regime is quite different from visible and infrared frequencies, so the scale change cannot be applied as easily.

Authors Response:

We agree with the referee on this point. Our intention is that the design principle and pre-trained phase masks can be scaled into various frequencies. If material dispersion is considered, we should design specific metamaterials in experiment to match the phase masks for different frequencies (e.g., microwave, terahertz, optical). These metamaterials are distinct, such as TiO_2 metasurfaces in visible [arXiv:2107.07873] and metallic resonant metasurfaces in microwave [Adv. Opt. Mater. 5, 1600506 (2017)]. To avoid possible misunderstanding, we have re-organized the sentences on lines 152-154 in the new submission.

"This design principle and pre-trained phase masks can be scaled into other single frequency. For different frequencies, we should design specific metamaterials due to material dispersions to match the pre-trained phase masks, such as TiO_2 metasurfaces in visible [35]."

Referee #2 -- Comment 8:

- On line 219, it is mentioned that the experiment achieves 98% accuracy utilizing layered neuro-metamaterials for identifying the rabbits' postures. It should be clearly explained how the quantitative accuracy was obtained in the experiment. I assume the evaluation was done by recording the video of the rabbits and comparing with its prediction signal. However, if that's the case, it could have been evaluated only on certain, selected frames, which is not appropriate. Thus, the method of calculating the experimental accuracy should be specified.

Authors Response:

We appreciate the referee for the good comment and apologize for the unclear meaning this sentence may cause. The 98% accuracy actually refers to the accuracy of test set in training, rather than in experiment; the detailed training process can be found in the main text. To experimentally validate the neuro-metamaterials, in principle, we should consider a large number of living rabbits to measure the output case-by-case and then calculate the average accuracy. However, it is extremely cumbersome. For simplicity, we consider two living rabbits play freely (lasting 10 seconds) in front of the neuro-metamaterials. The experimental results are summarized in Fig. 3 of the main text and two supplementary movies. Probably, it is difficult to give the referee a definite experimental accuracy because only two living rabbits were considered. However, the time-varying signals in Fig. 3 are in general consistence with the prediction, which gives readers a qualitative and intuitive evidence for the preliminary metrics of the neuro-metamaterials. Another point we want to emphasize is that, the experimental results are obtained based on two freely-playing rabbits without human intervention. This unmanned procedure can strongly verify our conclusion.

In the new submission, we have re-organized the sentences to clarify the 98% accuracy and how we did in experiment.

On lines 75-76, "After the neural network has been trained, the recognition accuracy rate reaches 98% in simulation for representative rabbit postures, including walking, standing, and sitting."

On lines 230-232, "In the experiment, we utilized layered neuro-metamaterials for identifying the postures of two freely-playing rabbits without human intervention, which can strongly verify the preliminary metrics of the neuro-metamaterials."

Referee #2 -- Comment 9:

- I wonder if these are typos:

In line 227, "Geforce '249 10' GTX"

In line 228, "a Linux '250' operating"

'249 10' and '250' are also written in the supplementary.

Authors Response:

Thanks for the careful reading. We have corrected the typos in the main text and supplementary materials.

On lines 257-258, "GeForce GTX TITAN X GPU and Intel(R) Xeon(R) CPU X5570 @2.93GHz with 48GB RAM, running a Linux operating system."

General comments from Referee #3:

Thanks for the revisions made by the authors in the last round in light of the referees' reports. However, my major concern is in the novelty of this work over the existing ones, which have not yet been well addressed in the revised manuscript. Thus, unfortunately, I cannot recommend its publication in Nature Communications in its current form. The main issues are identified below:

Authors Response:

We appreciate the referee for his/her patience and the constructive comments. In the following, we answer the specific questions point-by-point and clarify the novelty in **Referee #3 -- Comment 1 (continued)**.

Specific comments from Referee #3:

Referee #3 -- Comment 1:

1. In the authors' reply to my comments 5 and 7, their strategy applies to EM waves of different frequencies, while the metasurfaces designed based on it only works for a single frequency, which makes their statements of broadband functionality less convincing.

Authors Response:

We apologize for the possible confusion caused in the last response. The meaning we want to convey is that the design principle and pre-trained phase masks can be scaled into other single frequency (not broadband). For example, our microwave neuro-metamaterials (working at a single frequency) can be scaled into optical single frequency using TiO₂ metasurfaces [arXiv:2107.07873]. For broadband functionality, it is actually an open challenge, because it involves the dispersion of metamaterials and sophisticated optimizations. Recently, there are some works on the design of broadband metasurfaces (e.g., *Adv. Opt. Mater.* 9, 2001311 (2020)), which may provide feasible solutions to realize broadband dynamic recognition in the future.

In the new version, we have rephrased the statements on lines 152-154 to avoid possible misunderstanding.

"This design principle and pre-trained phase masks can be scaled into other single frequency. For different frequencies, we should design specific metamaterials due to material dispersions to match the pre-trained phase masks, such as TiO₂ metasurfaces in visible [35]."

Referee #3 -- Comment 1 (continued):

*Actually, their mechanism shows high similarity to the ones found in the literatures (e.g., *Science* 361, 1004-1008 (2018), *Photon. Res.* 7, 823-827 (2019) and *Nat. Comm.* 11, 6309 (2020) which respectively proposed the architecture of optical neuro-metamaterial described here, the optical artificial neural inference with dynamic effects and the metasurface-based passive neural network with functionality of recognizing object's shape).*

Authors Response:

We appreciate the referee for introducing these important references to us. We are also very familiar with these optical computing works and other related works enabled by metamaterials and photonics. In the mentioned works, [*Science* 361, 1004–1008 (2018)] demonstrates the concept of diffractive neural network and applies it for classification and imaging tasks at terahertz wave. [*Photon. Res.* 7, 823–827 (2019)]

theoretically shows a nanophotonic medium empowered artificial neural network and its classification capability. [*Nat. Comm.* 11, 6309 (2020)] shows an acoustic diffractive neural network and its applications in recognition and discerning misaligned orbital-angular-momentum vortices. These works greatly enrich the modality and application of optical computing, which have also been cited (Refs. [16,20,35]) in our work.

Compared with these works, our work shows three main novelties.

First, our work presents the ***first dynamic entirely-optical 3D object recognition***, while the mentioned and other relevant works are for 2D object. Practical objects are always 3D and much complicated than 2D. Therefore, we directly utilize the scattered fields of 3D object for processing. As implied in the main text, our work maybe not viewed best as a brand-new physical discovery of wave-based computing, but rather, as a big advance for a large class of passive optical processing components for real-time task-solving purpose.

Second, we emphasize that none of the works in the community thus far have come close to such a real-world and practical experiment, where two living rabbits play freely without any human intervention. This ***completely unmanned procedure*** provides strong evidences for the satisfactory performance of neuro-metamaterials. We believe that our work consists in this sense much more than a proof-of-concept demonstration or a pure theoretical demonstration, and thus brings optical computing more close to practical applications.

Third, we introduce and experimentally demonstrate ***a novel optical illusion mechanism***. Optical illusion was firstly proposed under transformation optics framework, which becomes the mainstream approach in this community [*Phys. Rev. Lett.* 102, 253902 (2009)]. However, it suffers from some fundamental limitations about the extreme parameter requirements (inhomogeneous and anisotropic) and stationary object, making it very difficult to be applied in practice. Our illusion mechanism substantially simplifies the experimental realization, with the big advantages of practical feasibility and non-stationary object. Here it is worth noting that our dynamic mirage is different from the purely theoretical work [*Photon. Res.* 7, 823–827 (2019)], where the dynamics are mainly about field-evolutions in classification process.

In the main text, we have added some sentences to clarify the main contributions of our work (**all above three novelties have also been clearly pointed out by Referee #2** on page R1). We hope it will be helpful for this referee to better understand our work.

On lines 64-65, “...to facilitate entirely optical object recognition for real-world three-dimensional (3D) applications”

On lines 234-235, “This novel scheme substantially simplifies the mainstream yet difficult-to-reach transformation optics-based optical illusion, because of its practical feasibility and dynamic input.”

On lines 230-232, “In the experiment, we utilized layered neuro-metamaterials for identifying the postures of two freely-playing rabbits without human intervention, which can strongly verify the preliminary metrics of the neuro-metamaterials.”

Referee #3 -- Comment 2:

2. In their reply to my comment 6, the method for producing dynamic illusion is innately identical to the imaging effect demonstrated in the previous work (see the Supplemental Materials of *Science* 361, 1004-1008 (2018))

which refers to a correspondence between the input and output images and imaging quality of illusion. What is more, in the current work the input and output data only include 15 images, which is far less than what was demonstrated there. Obviously, here the generalization ability of a neural network cannot be verified when the training and testing databases are same.

Authors Response:

Thanks for the good questions. We would like to explain that illusion is **different** from imaging. For imaging, its physical essence is to make a limited number of point sources at input plane to focus at output plane (see Fig. R2). To this end, [Science 361, 1004–1008 (2018)] created an imaging lens consisted of five-layer diffractive neural networks, each of which has 300×300 neurons. In training process, no matter what kind of database were used (ImageNet in that work or others), **the physical essence does not change**. And once trained well, the imaging lens can be naturally applied to arbitrary object (generalization ability noted by the referee) with a certain resolution.

For illusion, **it is almost impossible to be general like imaging lens, because illusion is site-specific**, for example, converting a moving rabbit into a moving giraffe, cat, and tiger. For different scenarios and users, the input and output may be different. It is almost impossible to construct an illusive neuro-metamaterials that can convert any input into the same output, or the same input into any output. In our work, to demonstrate the dynamic illusion concept, we only use a small-scale network ($70 \times 84 \times 2$) to convert a sequence of rabbit movements (15 images) into a different holographic video (15 images). If necessary, the number of images can be reasonably increased with a large-scale and more sophisticated neural network.

Figure R2 | Amplitude Imaging principle. (a,b) Amplitude and phase information of the wave that is propagating within and without multi-layered diffractive neural network. Note: this figure is copied from [Science 361, 1004–1008 (2018)].

In the new submission, we have added the above discussion on lines 215-221.

“We note that the optical mirage demonstrated here is different from imaging [16]. For imaging, its physical essence is to make a limited number of point sources at input plane to focus at output plane. Once designed well, the imaging lens can be naturally applied to arbitrary object (strong generalization ability) with a certain

resolution. For optical mirage, it is almost impossible to be general like that, because optical mirage is site-specific. For different scenarios and users, the input and output may be different, for example, converting a moving rabbit into a moving giraffe, cat, and tiger.”

Referee #3 -- Comment 3:

3. *The authors have not clarified the questions about the fundamental limitations on the optical dynamic illusions I raised in the last round. Please demonstrate the relationship between the frame rate and accuracy of illusion based on the current framework.*

Authors Response:

We apologize for missing this question in the last round. One of the fundamental limitations on the optical dynamic illusions may be the generalization (capacity). As clarified in the **Referee #3 -- Comment 2**, the dynamic illusion is site-specific and it is very challenging to be extended into a very general application. Actually, this is also an open challenge for state-of-the-art optical computing works [*Nature* 588, 39–47 (2020)].

Regarding “*the relationship between the frame rate and accuracy of illusion*”, we guess the referee means to the relationship between the number of input images (equivalent to frame rate) and accuracy of illusion. For a period of time, if the frame rate is higher, the number of input images is smaller. In this vein, we make some simulations and utilize structural similarity (SSIM) index to characterize the illusion performance [*IEEE Trans. Image Process.* 13, 600–612 (2004)]. Table 1 shows the simulated results. Obviously, for the same network framework, if the number of input images is smaller, the average SSIM index is higher.

Table 1 Relationship between the number of input images/frame rate and accuracy.

Input images	5	10	15
Frame rate	2.33 fps	4.66 fps	7 fps
SSIM	95.98%	94.62%	93.51%

In the new submission, we have added the above discussion.

On lines 205-206, “To quantitatively characterize the mirage performance, we adopt a structural similarity (SSIM) index with an average of 93.51% [38].”

On lines 221-222, “For a certain network, when the number of input images increase, the SSIM of output images may decrease.”

REVIEWER COMMENTS

Reviewer #2 (Remarks to the Author):

I thank the authors for revising the manuscript reflecting the comments and clarifying the questions/suggestions. During the revision twice, I believe the manuscript has improved a lot. Although there is still debating about the novelty issue, the result shown here already has enough scientific merit and impact for the broader community of optics, computer science, nanoscience and engineering. I recommend this manuscript to be published in Nature Communications.

Reviewer #3 (Remarks to the Author):

In the new version of manuscript, the authors have made substantial revisions that have addressed many questions raised by the referees. I need to point out, however, that the authors have misunderstood some of my concerns on the advantages of their scheme comparing with the previous works, which are crucial for judging the impact of this work.

1. In my previous Comment 2, by the generalisation ability of a neural network, I am not requiring the authors to be able to convert all the input objects into the same one based on the model they have in hand. Given that the authors claim that they achieved 'dynamic illusion' with important application potential, it would be natural in the context of real-world applications to ask the question of whether their scheme still works as the movement of rabbit is beyond the scope of these 15 static images (e.g., what if the head of rabbit raises higher? Will it result in a lower posture of the giraffe's head?) Such generalisation ability is significant for the application of the proposed scheme in practice. Notice that I am still following the authors' procedure of converting a moving rabbit into a moving giraffe. The problem is that the input and output of 15 static images alone is not sufficient for supporting their claimed 'dynamic illusion'. Rather, based on the results they already have I would still refer to it as a static holographic production that establishes a mapping between two groups of static images, except for some relationship between the images in the same group.

2. This is also the very propose of my Comment 3 suggesting the authors to further discuss the fundamental limitation of their 'illusion' mechanism. It would be helpful for readers to clarify the maximal length of input sequence their scheme supports when maintaining a relatively high structural similarity (SSIM) index. Is the 'illusion' functionality limited within a few of trained images, or does it also work for some untrained movements?

3. In addition, although the authors argue that the imaging and illusion mechanisms are different, the implementation of the illusion shown in this paper is the same as the implementation of imaging in Ref. [16], where the input and output images are set and the neural network is trained to minimise the loss of the network. The illusion mechanism proposed here is just an application of the neural network shown there.

Response Letter to Reviewers

We are grateful for the constructive comments on this manuscript (NCOMMS-21-15025B) from all the referees. In the text below, each comment is quoted in *italics* and is followed by the corresponding detailed response. We have also revised the manuscript and supplementary material accordingly. These updates are highlighted in blue and by a vertical red line in the left margin in those files. In the text below, the references to these updates are highlighted in a similar way (i.e., by a vertical red line).

General comments from Referee #2:

I thank the authors for revising the manuscript reflecting the comments and clarifying the questions/suggestions. During the revision twice, I believe the manuscript has improved a lot. Although there is still debating about the novelty issue, the result shown here already has enough scientific merit and impact for the broader community of optics, computer science, nanoscience and engineering. I recommend this manuscript to be published in Nature Communications.

Authors Response:

We thank the referee for the positive comments and the recommendation of our work.

General comments from Referee #3:

In the new version of manuscript, the authors have made substantial revisions that have addressed many questions raised by the referees. I need to point out, however, that the authors have misunderstood some of my concerns on the advantages of their scheme comparing with the previous works, which are crucial for judging the impact of this work.

Authors Response:

We appreciate the referee for his/her careful reading and acknowledgement of our efforts. In the following, we answer the remaining questions point-by-point.

Specific comments from Referee #3:

Referee #3 -- Comment 1:

1. In my previous Comment 2, by the generalisation ability of a neural network, I am not requiring the authors to be able to convert all the input objects into the same one based on the model they have in hand. Given that the authors claim that they achieved 'dynamic illusion' with important application potential, it would be natural in the context of real-world applications to ask the question of whether their scheme still works as the movement of rabbit is beyond the scope of these 15 static images (e.g., what if the head of rabbit raises higher? Will it result in a lower posture of the giraffe's head?) Such generalisation ability is significant for the application of the proposed scheme in practice. Notice that I am still following the authors' procedure of converting a moving rabbit into a moving giraffe. The problem is that the input and output of 15 static images alone is not sufficient for supporting their claimed 'dynamic illusion'. Rather, based on the results they already have I would

still refer to it as a static holographic production that establishes a mapping between two groups of static images, except for some relationship between the images in the same group.

Authors Response:

We thank the referee for the very good question. The first point we would like to elucidate is that neuro-metamaterials can be applied for inference-based and optimization-based tasks, both of which are useful for different application demands. In our work, we demonstrate the two capabilities via posture recognition and optical illusion experiments, respectively.

Optical illusion here is treated as an optimization task, not an inference task. It implies that the images distinct from or outside the pre-defined optimization range (15 images) may not work well. This treatment is similar to the pioneering illusion work [*Phys. Rev. Lett.* 102, 253902 (2009)]. Although the number of input images is small in the proof-of-the-concept experiment, we can still observe the continuous movement of a rabbit and a giraffe; this presented dynamic illusion actually outperforms most of previous illusions working for a single input [e.g., *Adv. Mater.* 27, 4628-4633 (2015) & *Phys. Rev. B* 101, 024104 (2020)]. Towards real-world applications, one can incorporate more desired images in advance. As shown in Fig. R1, neuro-metamaterials show a powerful capacity to implant a freely-playing rabbit (80 images) to a dynamic giraffe mirage.

On the other hand, to dispel the referee's concern about the generalization ability, we can also deal with optical illusion as an inference (machine learning) task using neuro-metamaterials. Such generalization capability actually has been well verified in our first posture recognition experiment. To validate this again, we carry out another illusion experiment to transform rabbit (with different postures of walking, sitting, and standing) into a drinking giraffe with different postures based on the dataset in the first posture recognition experiment. After an adequate training, the neuro-metamaterials can work for untrained images to produce a dynamic illusion (Fig. R2). Even when the rabbit's head raises higher, the output giraffe's head goes down.

Fig. R1 | Dynamic illusion example that incorporates 80 images as an optimization task. a, Snapshots of the moving rabbit. b, Output mirage. Compared with the demonstration in the main text, this example incorporates more images, and exhibits a much clearer and huger movement.

Fig. R2 | Illusion results for untrained input as an inference task. In this experiment, we treat dynamic illusion as an inference task, aiming to transform rabbit (with different postures of walking, sitting, and standing) into a drinking giraffe. The five-pointed stars represent the locations of the giraffe’s head and the dotted line is added as a reference. When the rabbit’s head raised higher, the giraffe’s head goes down gradually. For ⑥, the rabbit’s head is the highest and does not appear among the training samples.

In the new submission, we have added Fig. R1 as Fig. S5, Fig. R2 as Fig. S6, and the above discussion,

On lines 197-198, “Notice that optical mirage here is treated as an optimization task, rather than an inference task (which has been verified in the posture recognition experiment).”

On line 221, “optical mirage is site-specific for a given input/output sequence or some input/output categories.”

On lines 223-225, “In real-world applications, we may reasonably incorporate more images into large-scale neuro-metamaterials as an optimization task (Fig. S5), or treat optical mirage as an inference task for untrained input (Fig. S6).”

Referee #3 -- Comment 2:

2. This is also the very propose of my Comment 3 suggesting the authors to further discuss the fundamental limitation of their ‘illusion’ mechanism. It would be helpful for readers to clarify the maximal length of input sequence their scheme supports when maintaining a relatively high structural similarity (SSIM) index. Is the ‘illusion’ functionality limited within a few of trained images, or does it also work for some untrained movements?

Authors Response:

Thanks for the helpful suggestion. The maximal length of input sequence depends on the specific network framework and input data. Here, we take the small-scale network in the main text and an experimentally recorded movie as example to evaluate the maximal length. The result shows that the length of input sequence reaches about 150 images with an average SSIM of 75%, which can guarantee a relatively satisfactory illusion.

According to the above analysis, the illusion functionality can be applicable for hundreds of trained images with a small-scale network. With a more sophisticated network configuration, the number will increase greatly [Science 361, 1004-1008 (2018)]. For untrained movements, the optical illusion may not work well in our case, because here it is an optimization task, not an inference task; please refer to the response to Comment #1.

In the new submission, we have added the above discussion on lines 226-228,

“Taking the small-scale network and an experimentally recorded movie as example, the length of input sequence reaches about 150 images with an average SSIM of 75%.”

Referee #3 -- Comment 3:

3. In addition, although the authors argue that the imaging and illusion mechanisms are different, the implementation of the illusion shown in this paper is the same as the implementation of imaging in Ref. [16], where the input and output images are set and the neural network is trained to minimise the loss of the network. The illusion mechanism proposed here is just an application of the neural network shown there.

Authors Response:

We thank the referee for pointing this out. In a more general view, we agree with the referee that “*the implementation of the illusion...*”, or understand illusion as a special case of imaging. Both imaging and illusion utilize optical analogy computing technique to manipulate the scattered fields at will, and can be summarized as the minimization of the loss of the network. Actually, this is also a universal procedure for a majority of optical analogy computing related works.

On the other hand, we would like to clarify their differences in the following view. For imaging, the working principle is to make a limited number of point sources to focus at output plane. It is very natural for a well-trained imaging lens to be applied for arbitrary object. While for illusion, the working principle is not to make light focus, but with complicated interference to construct different images. Because the output depends on different scenarios, the illusion cannot be applied for arbitrary object and very universal as imaging lens.

We hope the above explanation could help the referee better understand illusion.

REVIEWER COMMENTS

Reviewer #3 (Remarks to the Author):

I appreciate the revisions made by the authors which manifest they did a great effort in addressing my concerns. However, there are still two concerns remained to be addressed before the manuscript can be considered for publication.

In the response to Comment 1, the authors claim that the optical illusion here is treated as an optimisation task instead of an inference task. It is obvious however, that the light field in different images demonstrated in their results bears very strong correlation (Take for instance the giraffe images which are almost identical except for the locations of head and neck). In other words, there is a strong similarity between the sub-tasks. In this context, it would be natural to ask a question whether or not the performance of their proposed mechanism would deteriorate as the similarity between sub-tasks reduces, e.g., as the standing-up of a rabbit is mapped to the change in the whole body's posture of giraffe.

In their response to the Comment 2 the authors increase the length of the input sequence by simply increasing the sampling ratio of the dynamic video while keeping the first and last frames identical with the sequence in the original version of manuscript. This does increase the lengths of the input and output sequences, but at the same time results in a much higher correlation between two neighboring images (viz., the similarity between sub-tasks grows despite the seeming increase in the number of tasks their neural network can handle). In order to better evaluate the fundamental limitations in their purpose scheme, I suggest the authors to also consider a new sequence for which the first 15 frames are chosen as the 15 images used in the original manuscript and the rest frames are designed as per the updated mapping between the rabbit and giraffe's postures. In such a case, what is the maximum sequence length they can input when ensuring an average SSIM exceeding 75%?

Response Letter to Reviewers

We are grateful for the constructive comments on this manuscript (NCOMMS-21-15025C) from all the referees. In the text below, each comment is quoted in *italics* and is followed by the corresponding detailed response. We have also revised the manuscript and supplementary material accordingly. These updates are highlighted in blue and by a vertical red line in the left margin in those files. In the text below, the references to these updates are highlighted in a similar way (i.e., by a vertical red line).

General comments from Referee #3:

I appreciate the revisions made by the authors which manifest they did a great effort in addressing my concerns. However, there are still two concerns remained to be addressed before the manuscript can be considered for publication.

Authors Response:

We appreciate the referee for the positive recognition of our efforts and the valuable suggestions. In the following, we answer the remaining two concerns point-by-point.

Specific comments from Referee #3:

Referee #3 -- Comment 1:

In the response to Comment 1, the authors claim that the optical illusion here is treated as an optimisation task instead of an inference task. It is obvious however, that the light field in different images demonstrated in their results bears very strong correlation (Take for instance the giraffe images which are almost identical except for the locations of head and neck). In other words, there is a strong similarity between the sub-tasks. In this context, it would be natural to ask a question whether or not the performance of their proposed mechanism would deteriorate as the similarity between sub-tasks reduces, e.g., as the standing-up of a rabbit is mapped to the change in the whole body's posture of giraffe.

Authors Response:

Thanks for the insightful question. We have followed the referee's suggestion to conduct another experiment, in which the output images distinct from each other (a weak similarity among the sub-tasks). As shown in Fig. R1, optical illusion can still work well for a running giraffe (the change in the whole body's posture of giraffe). In the new submission, we have incorporated Fig. R1 as Fig. S6, and the related description in the main text.

On lines 225-227, "The result in Fig. S6 shows that the optical illusion can still work for a running giraffe (for the case where the similarities among the output images are very weak)."

Fig. R1 | Dynamic illusion example of a running giraffe. a, Snapshots of the moving rabbit. **b,** Output mirage. Compared with the previous demonstrations, the similarity among the sub-tasks are weak.

Referee #3 -- Comment 2:

In their response to the Comment 2 the authors increase the length of the input sequence by simply increasing the sampling ratio of the dynamic video while keeping the first and last frames identical with the sequence in the original version of manuscript. This does increase the lengths of the input and output sequences, but at the same time results in a much higher correlation between two neighboring images (viz., the similarity between sub-tasks grows despite the seeming increase in the number of tasks their neural network can handle). In order to better evaluate the fundamental limitations in their purpose scheme, I suggest the authors to also consider a new sequence for which the first 15 frames are chosen as the 15 images used in the original manuscript and the rest frames are designed as per the updated mapping between the rabbit and giraffe's postures. In such a case, what is the maximum sequence length they can input when ensuring an average SSIM exceeding 75%?

Authors Response:

We thank the referee for the comment. As suggested, we fed the 15 images used in the original manuscript and the updated mapping between the rabbit and giraffe's postures into the neuro-metamaterials to re-examine the maximal length. We conclude that the length of input sequence reaches about 130 images with an average SSIM of 75%, which is comparable to that (150 images) in the last response.

REVIEWERS' COMMENTS

Reviewer #3 (Remarks to the Author):

The authors have made satisfying revisions and I think the manuscript is ready for publication in its present form.

Response Letter to Reviewers

We are grateful for the constructive comments on this manuscript (NCOMMS-21-15025D) from all the referees. These comments are very valuable and helpful for improving our manuscript. In the text below, each comment is quoted in *italics* and is followed by the corresponding response.

Comments from Referee #3:

The authors have made satisfying revisions and I think the manuscript is ready for publication in its present form.

Authors Response:

We thank the referee for the positive comments and the recommendation of our work.